Physiological, nutritional, and molecular responses of Brazilian sugarcane cultivars under stress by aluminum

Oliveira Mariane de Souza 1
http://orcid.org/0000-0001-7014-012X Rocha Sâmara Vieira 1 samara.vieira@hotmail.com
Schneider Vanessa Karine 1
Henrique-Silva Flavio 1
Soares Marcio Roberto 2
Soares-Costa Andrea 1 andreasc@ufscar.br
1 Department of Genetics and Evolution, Federal University of São Carlos , São Carlos, SP , Brazil
2 Department of Natural Resources and Environmental Protection/Agrarian Sciences Center, Federal University of São Carlos , Araras, SP , Brazil
Ribeiro-Barros Ana
Electronic publication date: 2021 Jun 28
Publication date: 2021
Volume: 9
Electronic Location ID: e11461
Received 2020 Oct 23; Accepted 2021 Apr 26
Copyright: © 2021 Oliveira et al.
Copyright year: 2021
Copyright holder: Oliveira et al.
License: This is an open access article distributed under the terms of the Creative Commons Attribution License, which permits unrestricted use, distribution, reproduction and adaptation in any medium and for any purpose provided that it is properly attributed. For attribution, the original author(s), title, publication source (PeerJ) and either DOI or URL of the article must be cited.
License URL: https://creativecommons.org/licenses/by/4.0/

Keywords: Acid soil, Abiotic stress, Photosynthesis, Nutrients, Sugarcane, Aluminum, Al stress

Funding: FAPESP 2013.06318-2 and 2013.05370-0 This work was supported by the FAPESP (Foundation for Research Support of the State of São Paulo) process number 2013.06318-2 and 2013.05370-0 for supporting the research. The funders had no role in study design, data collection and analysis, decision to publish, or preparation of the manuscript.

==============================
Background

Sugarcane is a crop of global importance and has been expanding to areas with soils containing high levels of exchangeable aluminum (Al), which is a limiting factor for crop development in acidic soils. The study of the sugarcane physiological and nutritional behavior together with patterns of gene expression in response to Al stress may provide a basis for effective strategies to increase crop productivity in acidic soils.

Methods

Sugarcane cultivars were evaluated for physiological parameters (photosynthesis, stomatal conductance, and transpiration), nutrient (N, P, K, Ca, Mg, and S) and Al contents in leaves and roots and gene expression, of the genes MDH, SDH by qPCR, both related to the production of organic acids, and SOD, related to oxidative stress.

Results

Brazilian sugarcane RB867515, RB928064, and RB935744 cultivars exhibited very different responses to induced stress by Al. Exposure to Al caused up-regulation (SOD and MDH) or down-regulation (SDH, MDH, and SOD), depending on the cultivar, Al level, and plant tissue. The RB867515 cultivar was the most Al-tolerant, showing no decline of nutrient content in plant tissue, photosynthesis, transpiration, and stomatal conductance after exposure to Al; it exhibited the highest Al content in the roots, and showed important MDH and SOD gene expression in the roots. RB928064 only showed low expression of SOD in roots and leaves, while RB935744 showed important expression of the SOD gene only in the leaves. Sugarcane cultivars were classified in the following descending Al-tolerance order: RB867515 > RB928064 = RB935744. These results may contribute to the obtention of Al-tolerant cultivars that can play their genetic potential in soils of low fertility and with low demand for agricultural inputs; the selection of potential plants for breeding programs; the elucidation of Al detoxification mechanisms employed by sugarcane cultivars.

Introduction

Sugarcane crop has global economic importance with a harvested area of 27.6 million hectares, and a 2-billion-ton production. Brazil is the largest producer followed by India, China, and Thailand (FAO, 2018). It is estimated that 630.7 million tons will be harvested in the 2020/21 Brazilian harvest seasons (Conab, 2020).

Sugarcane plantations are often criticized for occupying large extensions of fertile arable land that could be used for food production. As a C4 plant, sugarcane has high photosynthetic efficiency (Sales et al., 2018). It accumulates large biomass under tropical climate conditions, with a high potential for cultivation on non-favorable agricultural land. Establishing sugarcane plantations on marginal and degraded lands is an important strategy to avoid land-use change and competition for areas used for food production or environmental preservation (Furtado et al., 2014; Bordonal et al., 2018).

Sugarcane has been mainly planted in the Southeast region of Brazil, close to most sugarcane mills. Since 2005, sugarcane cropping started expanding fast towards Central Brazil, a region that is dominated by the Cerrado biome (Adami et al., 2012; Filoso et al., 2015; Arruda, Giller & Slingerland, 2017). Cerrado is a savanna vegetation type that is generally flat with many microelevations (Arruda, Giller & Slingerland, 2017). It is ideal for mechanization and suitable for large-scale cropping and cattle-raising due to its soil and climate. Soil fertility is among the main biotic and abiotic factors influencing the productivity and technological quality of sugarcane (Viana et al., 2020). Soils from Central Brazil present high acidity, high levels of exchangeable aluminum (Al3+), and a general deficiency of nutrients (mainly phosphorus, calcium, and magnesium) (Yamada, 2005). Acid soils comprise large agricultural areas in tropical and subtropical regions, which comprises approximately 50% of the arable land (Sade et al., 2016; Singh et al., 2017).

Soil Al solubility increases at low pH levels (mainly at pHH2O ≤ 5.0), and its toxicity is a limiting factor for crop development in acidic soils (Ryan, 2018). The most sensitive plant region to Al is the root apex (Gupta, Gaurav & Kumar, 2013; Rao et al., 2016; Ryan, 2018; Yadav et al., 2020). The cation Al3+ binds to the cell wall, particularly in the pectin matrix, negatively charged with carboxylic groups; it also binds to the apoplastic side of the plasma membrane, impairing its function (Horst, Wang & Eticha, 2010). The cation interferes with cell division and root growth, reducing DNA replication since the double helix ribbon presents increased stiffness. Aluminum toxicity alters enzyme mechanisms, results in polysaccharide accumulation, affects cellular respiration, and modifies the structure and function of the plasma membrane, which impairs the absorption of water and nutrients (Zheng, 2010; Patra et al., 2020). Effects of Al on shoot growth may be a secondary consequence that includes a reduction of photosynthesis (Vitorello, Capaldi & Stefanuto, 2005; Ying & Liu, 2005; Mihailovic, Drazic & Vucinic, 2008; Lazarević, Horvat & Poljak, 2014). Aluminum may hinder chloroplast formation and modify its function (Moustakas, Eleftheriou & Ouzounidou, 1997). As a result, there is a decrease in stomatal conductance and biochemical reactions of CO2 fixation (Vitorello, Capaldi & Stefanuto, 2005; Mukhopadyay et al., 2012).

Biotechnological approaches have contributed to understanding the effects of Al in plants along with the function and interaction of resistance genes (Kochian et al., 2015; Ryan, 2018; Patra et al., 2020). Aluminum induces the expression of different genes, like TaALMT1 in wheat, an Al-activated malate transporter (Sasaki et al., 2004); AtMATE, an Al-activated citrate transporter in Arabidopsis (Liu et al., 2009); ALS3 that redistributes Al accumulated away from sensitive tissues to protect the growing root from the toxic effects of Al; ALS1 in Arabidopsis that encodes a root tip and stele localized half type ABC transporter required for root growth in Al toxic environments (Larsen et al., 2005; Larsen et al., 2007); STAR1 and STAR2 that form a complex that functions as an ABC transporter, which is required for detoxification of Al in rice (Huang et al., 2009); SbMATE, an Al-activated citrate transporter in sorghum (Magalhães et al., 2007); HvAACT1 that secretes citrate to the rhizosphere in barley (Fujii et al., 2012); and ScFRDL2, involved in Al-activated citrate secretion in the rye (Yokosho, Yamaji & Ma, 2010). There are several Al tolerance mechanisms involved in different biochemical pathways with one or more genes. They differ among species and even among cultivars of the same species. This difference is expected since plants may differ in Al mobilization capacity, absorption rate, nutritional requirements, Al resistance, and nutrient balance maintenance in acidic soils (Poschenrieder et al., 2008).

The quantitative molecular technique of qPCR has been widely used for gene expression analysis in Al tolerance studies. Gene expression response to Al was analyzed in maize regarding Zm.16676.1.A1 (putative cytochrome P450 superfamily protein), related to the control of both biosynthesis and inactivation of gibberellin; Zm.3634.1.A1 (jasmonate-induced protein), involved in jasmonate biosynthesis; Zm.6272.1.A1 (UDP-glycosyltransferase), involved in hormone biosynthesis, metabolism of secondary metabolite, and responses to stress and xenobiotics; and Zm.1228.2.A1 (PHG1A protein), involved in cellular adhesion in higher eukaryotes (Mattiello et al., 2014). Genome-wide transcriptomic analyses have been used to examine buckwheat (Fagopyrum tataricum) for Al-tolerance mechanisms (Zhu et al., 2015). Studies have also looked at the ASR1 and ASR5 genes in rice, as well as complementary transcription factors that regulate gene expression in response to Al (Arenhart et al., 2016).

Several genes are differentially regulated by Al stress in different plant species. In sugarcane, histone deacetylase (P56521), serine/threonine kinase (AP002482), and RAS-related protein RGP1 (P25766) have been described to have different expression pattern in presence of Al (Watt, 2003). Drummond et al. (2001) identified several genes with high similarity, including Superoxide dismutase Cu-Zn, and Phospholipid hydroperoxide glutathione peroxidase-like. These genes code enzymes that alleviate oxidative stress, or combat infection by pathogens like War13.2–oxalate oxidase and wali4–phenylalanine ammonia-lyase, and genes that code for proteins responsible for the release of organic acids, Succinate-CoA ligase–beta chain and Malate dehydrogenase, and signal transducers, NtGDI1–GDP dissociation inhibitor and SLT2–MAP–kinase. All of these genes are related to SAS 43.141 (Sugarcane Assembled Sequences) in sugarcane, which are available at SUCEST (Sugarcane EST Project–http://sucest-fun.org/; Vettore et al., 2001), including genes related to Al tolerance described in the literature. Differentially regulated genes have been identified in two sugarcane cultivars with more than 4,000 differentially expressed genes (DEGs) in the tolerant cultivar and only about 850 DEGs in the Al sensitive cultivar. Among several transcripts, the ARF (Auxin responds factor), MT1 (metallothionein-like protein 1), CDPK (calcium-dependent protein kinase), and Glutathione S-transferases genes were substantially active in Al tolerance mechanisms; ARF was related to auxin signaling, MTI to detoxification, CDPK to signal transduction, and Glutathione S-transferases it ROS protection mechanism (Rosa-Santos et al., 2020).

Sugarcane possesses one of the most complex plant genomes, presenting a level of polyploidy ranging from 5 to 16 (Hoang et al., 2015). The response of this crop to Al toxicity remains poorly understood ever since the pioneer study by Hetherington, Asher & Blamey (1988) compared Al tolerance of three Australian sugarcane cultivars, Q77, Q113, and Q117, in a short-term solution culture experiment. Studies on genes related to sugarcane tolerance to Al are scarce (Drummond et al., 2001; Watt, 2003). Gene expression patterns are determinants of physiological processes that can modify plants’ cellular properties associated with stress (Rickes et al., 2019; Rosa-Santos et al., 2020). Determining gene expression patterns in response to Al stress could improve our understanding of their functions so that we can establish effective strategies to cope with stress conditions.

In this study, we present a physiological, nutritional, and molecular evaluation of Brazilian sugarcane cultivars. We examined nutrient and Al content in roots and in leaves, and physiological parameters as transpiration rate, stomatal conductance, photosynthesis, and chlorophyll content. Moreover, we performed a gene expression analysis of succinate dehydrogenase (SDH), malate dehydrogenase (MDH), and superoxide dismutase (SOD).

Materials & methods

Plant material and growth conditions

Experiments were carried out at the Agrarian Sciences Center of the Federal University of São Carlos (UFSCar), Araras-SP, Brazil (22°21′ S lat; 47°23′ W long; 690 m altitude). According to Köppen’s climatological classification, the regional climate is Cwa type (called tropical highland), mesothermic with warm and humid summer and dry winter. The average annual precipitation is 1,430 mm and the average annual temperature is 21.5 °C.

Three sugarcane cultivars were selected: RB928064 (medium cycle; high soil fertility potential), RB867515 (medium cycle; medium to low soil fertility potential), and RB935744 (late cycle; medium soil fertility potential) (RIDESA, 2010). Sugarcane mini-stalks were provided by the Sugarcane Genetic Improvement Program at UFSCar, which is a part of the Inter-University Network for the Development of Sugar and Energy Sector. Sugarcane mini-stalks with one bud were cut to a length of 6 cm and heat-treated (52 °C for 20 min) to prevent chlorotic streak disease and stimulate germination. The setts were then dipped in fungicide (6.5% benomyl) for 30 s and sprouted in plastic trays with medium-texture vermiculite as a substrate when dry (water: vermiculite ratio 1:0.8 (wt/wt)). Stalk borders were previously sealed to avoid excessive oxidation and fungal contamination by dipping in hot paraffin wax (Hetherington, Asher & Blamey, 1988).

The mini-stalks were placed in trays with the bud side up and covered with a thin layer of vermiculite. Trays were placed in a germination chamber at a constant temperature of 30 °C and intermittent light photoperiod. The average pre-sprouting period was 15 d with some variation according to sugarcane cultivar. Substrate moisture was controlled daily with water replaced every 48 h. These steps were performed for both greenhouse and nutritive-solution experiments.

Greenhouse bioassay

Subsoil samples (0.2–0.4 m) of Dystrophic Typic Hapludox (Oxisol) were collected in a sugarcane cultivation area at Santa Lucia Sugar and Alcohol Mill (southeast region of Brazil; 22°18′S and 47°23′W). Both chemical and physical analysis were performed on air-dried soil samples sorted in a 2 mm mesh sieve, following official Brazilian procedure (Raij et al., 2001; Camargo et al., 2009) that is aligned with well-known international methods of soil analysis (Sparks et al., 1996; Dane & Topp, 2002). Soil pHCaCl2 was determined potentiometrically in 0.01 M CaCl2 (1:2.5 soil: solution ratio) using a combined glass calomel electrode. Soil organic matter was quantified by oxidation with potassium dichromate in the presence of sulfuric acid, followed by titration with ammonium Fe2+ sulfate (classical Walkley–Black method). P, K, Ca, and Mg content was extracted by the ion exchange resin method. P content was determined by photocolorimetry, K content by flame atomic emission spectrometry, and Ca and Mg content by atomic absorption spectroscopy (AAS). Exchangeable aluminum (Al3+) was extracted with 1 M KCl solution and determined by titration with 0.025 M NaOH. Soil potential acidity (H + Al) was extracted by 0.5 M calcium acetate solution at pH 7.0 and determined by titration with 0.025 M NaOH. Total cation exchange capacity (TCEC = SB + (H + Al)) was obtained as the sum of exchangeable bases (EB=Ca+Mg+K) and soil potential acidity. Base saturation was obtained by V% = (EB/TCEC) × 100, and aluminum saturation was calculated by m% = (Al/EB + Al) × 100. S content was extracted by 0.01 M CaH2PO4 solution and determined by turbidimetry. Boron content was extracted by hot water and determined by the azomethine-H method. Cu, Fe, Mn, and Zn content was extracted by DTPA-TEA solution at pH 7.3 and determined by AAS. Clay, sand, and silt content was determined by the pipette method and the moisture at field capacity (FC) was determined at tensions of 6, 10, and 33 kPa.

Soil samples were sorted into two treatments: a control group with the addition of limestone and an aluminum treatment group with aluminum chloride. Limestone was added at a dosage equivalent to 1.3 t ha−1 to a portion of the soil samples to increase the soil base saturation (V%) to 60% for the adequate cultivation of sugarcane (Raij et al., 1996). Aluminum chloride hexahydrate (AlCl3.6H2O-Synth®) was added to another portion of the soil samples to increase the soil Al saturation (m%) above 50% and induce Al stress. Soil samples incubated with limestone and those incubated with AlCl3.6H2O were sieved again after 30 days.

Soil columns were made by overlapping six rigid polyvinyl chloride (PVC) rings, which were attached externally using tape and special PVC adhesive. Each ring was 20 cm in diameter and 20 cm in height. The final height of each soil column was 1.2 m. The columns were filled with soil up to 1.5 cm from the top edge, forming a soil column of 118.5 cm in height and containing 37.8 dm3 of soil. The soil samples amended with limestone were used to fill the control columns and the first upper ring of the columns containing treatments with Al. This was done to provide conditions for good initial development of sugarcane stalks and to simulate the conditions of the arable layer (0.0–0.2 m) of cultivated soil.

After a pre-sprouting period, two stalks were transplanted into each soil column. At the end of one week, stalks that showed low development were removed. Each column initially received water pulses to maintain moisture and ensure the establishment of tillers. After the establishment period, soil moisture was maintained at 70% of field capacity, monitored by weighing. The experiment was conducted with a completely randomized design using five replicates for each treatment and cultivar, totalizing 30 columns.

Each column received plant fertilization (Raij et al., 1996) with 1.3 g of potassium chloride (58% of K2O), 2.35 g of simple superphosphate (17% of P2O5), and 0.24 g of ammonium sulfate (21% of N), based on chemical analysis of the soil (Table 1). In topdressing, fertilization was applied again using the same amount of K2O and N 30 days after the transplant. The experiment was conducted for six months under greenhouse conditions. Physiological parameters (transpiration rate (mmol H2O m−2s−1), photosynthesis rate (CO2 µmol m−2s−1), and stomatal conductance (H2O mol m−2s−1)) were analyzed using an infra-red gas analyzer (IRGA) with an airflow of 300 mL min−1 and a coupled light source of 995 mmol m−2s−1 (ADC, model LCi, Hoddesdon, UK). The measurement was performed on the third fully expanded leaf, in the morning of the last week of the experiment (between 8:00 am and 10:00 am) for all experimental plants (total of 30 plants).

Table 1 Chemical and physical attributes of subsoil samples from dystrophic typic hapludox.

Depth	PResin	OM	pH	K	Ca	Mg	H + Al	Al	EB	TCEC	V	
(m)	mg dm−3	g dm−3	CaCl2	-----------------mmolc dm−3-------------------	%	
0.2–0.4	4	7	4.2	0.2	7	3	24	4.1	10	34	29.4	
Depth	m%	S	B	Cu	Fe	Mn	Zn	Clay	Sand	Silt	FC	
(m)	%	---------------mg dm−3-----------	-----------g kg−1--------	m3 m−3	
0.2–0.4	29.1	57	0.01	0.3	14	1,7	0,6	160	790	50	0.30	
Note:

OM = organic matter; H + Al = soil potential acidity; EB = exchangeable bases; TCEC = total cation exchange capacity; V% = base saturation; m% = aluminum saturation; FC = moisture at field capacity.

At the end of the experiment, samples of plant tissue were collected to analyze nutrient and Al content in the leaves and roots, and for chlorophyll extraction. The last ring of each soil column was cut and washed to completely remove soil attached to the root system. The entire root volume of this ring was separated for further analysis. Leaf samples were collected according to the morphological description of the crop and the Kuijper system of leaf description. Representative leaf samples of each treatment were formed by the central third of leaves +1 and +2 without the ribs for all replicates. Samples of roots and leaves were wrapped in labeled paper bags for transport to a forced-air oven and dried at 65 °C. Milling was carried out in Willey mills with knives, a stainless steel chamber, and sieves of 1 mm in diameter to ensure homogenization of the sample. Plant tissue samples were submitted to chemical analysis according to Malavolta, Vitti & Oliveira (1997). Root and leaf samples were subjected to solubilization with concentrated sulfuric acid, and the N content was determined by the semi-micro Kjeldahl method. To determine the P, K, Ca, Mg, S, Cu, Fe, Mn, and Zn content, samples were first solubilized with nitric (65%) and perchloric acids (70%). The P content was determined by vanadium-yellow spectrometry. The K content was determined by emission flame spectrometry at wavelengths between 766 and 767 nm. The Ca, Mg, Cu, Fe, Mn, and Zn content was determined by AAS with specific hollow-cathode lamps for each element, and the S content was measured by turbidimetry using a UV-VIS spectrometer. The remaining samples were incinerated in an electric muffle furnace at a temperature between 500 °C and 550 °C, and the resulting ash was dissolved in a 0.1 M nitric acid solution. The resulting extract was used for the determination of B by the azomethine-H method. For Al determination, dry vegetable matter was also oxidized by incineration. Hydrochloric acid (0.1 M) was added to the burned material, and the formed extract was treated with aluminon (0.10%) (triammonium salt of aurintricarboxylic acid). Aluminon interacts with Al to form a complex, and the intensity of the developed color was measured using a spectrophotometer at 520 nm according to a standard curve with predetermined Al concentrations (Brauner, Catani & Bittencourt, 1966).

Extractions of chlorophyll a, chlorophyll b, and total chlorophyll (a + b) were performed using leaf fragment samples of the central third of diagnostic leaves +1. For the analysis, 0.5 g of plant tissue was macerated with 5 mL of 80% acetone and filtered under vacuum. The final volume was adjusted to 50 mL with 80% acetone. The extracts were measured in a spectrophotometer to the read absorbance at wavelengths of 663 nm (chlorophyll a), 645 nm (chlorophyll b), and 652 nm (total chlorophyll (a + b)). The chlorophyll content is expressed in µg g-1 fresh sample mass according to the following equations (Eqs. (1)–(3)) (Carlin, Rhein & Santos, 2012):

(1) Chlorophyllb=((22.9×(A645)–4.68×(A663))V1,000×W×1,000

(2) TotalChlorophyll(a+b)=(((A652)×1,000)34.5×V1,000×W×1,000

(3) Chlorophylla=((12.7×(A663–2.69×(A645))V1,000×W×1,000

A = absorbance (nm); V = final volume of the extract (80% acetone + chlorophyll); W = fresh sample mass (g).

The results were compared by analysis of variance (ANOVA), using the program StatSoft STATISTICA 7.0. Upon determining significance for the F test (p < 0.05), averages were compared using Tukey’s test (p < 0.05).

Nutritive solution analysis

Mini-stalks with vigor and homogeneity were selected from the trays, fixed in styrofoam plates, and transferred to black receptacles (34 cm × 53 cm × 13 cm; capacity of 23.5 L) containing 17 L of nutritive solution to avoid light exposure. Before applying Al stress, adequate mineral nutrition for the sugarcane mini-stalks was established using an adaptation of the classic solution from Hoagland and Arnon, which was prepared based on the foliar nutrient content considered suitable for sugarcane (Raij et al., 1996). Macro and micronutrient solutions were calculated for the sugarcane plants (Cometti et al., 2006), which had the following composition: (i) macronutrients (g L−1)–calcium nitrate (0.407), potassium nitrate (0.547), monoammonium phosphate (0.181), and magnesium sulfate (0.150); (ii) micronutrients (mg L−1)–boric acid (0.30), copper sulfate (0.15), iron sulfate (2.40), manganese sulfate (1.50), sodium molybdate (0.03), and zinc sulfate (0.15). The nutritive solution was used to guarantee full conditions of a balanced nutrient supply to the stalks, and especially the roots to prevent limitations that are not caused by Al stress.

Stalks were fixed on polystyrene boards for root system immersion. The set was maintained under aeration through a bubbling air motor compressor. Uninterrupted light incidence equivalent to 5000 lux was applied with a 40-W fluorescent light in FITOTRON® Plant Growth Chambers for 24 h. After six days of immersion in nutritive solution, 30 mini-stalks with root uniformity were selected, representing 5 replicates for each cultivar and treatment. The stalks were placed in the holes of a polystyrene board and transferred to rigid polyvinyl chloride receptacles with 10 cm diameter and 20 cm height. The receptacles were filled with 2.7 L of nutritive solution, and two different Al concentrations were applied: 0 (control) and 3,000 µmol L−1, which were provided using aluminum chloride hexahydrate (AlCl3.6H2O). The roots were exposed to Al solution for 6 days. The pH nutritive solution was adjusted daily to 4.0 (±0.2) with 0.1 mol L−1 HCl to ensure the predominance of trivalent free species of Al [Al(H2O)6³+] (Rossiello & Jacob Netto, 2006). Fe was omitted from the solution to avoid interference, and P concentration was reduced to 0.0025 mmol L−1 due to issues with Al precipitation (Braccini et al., 2000).

qPCR analysis

Genes that have well-known or putative functions in the mechanism of aluminum tolerance (Vettore et al., 2001; Casu et al., 2004) were chosen for gene expression analysis using quantitative PCR (qPCR) (Bustin, 2009). A total of Two independent experiments were performed to evaluate the gene expression of the SDH, MDH, and SOD genes (Table 2) in the roots and shoots of three sugarcane cultivars exposed to aluminum. These genes were chosen for being related to Al tolerance. They have been described for other graminaceous crops that present high similarity with sugarcane. Root apex and shoot segments were collected from plants after 6 days of exposure to aluminum or to nutritive solution alone (non-treatment group). They were then immediately frozen in liquid nitrogen. Plant material was disrupted in liquid nitrogen with a mortar and pestle, and RNA was isolated using TRIzol reagent (Invitrogen) according to the manufacturer’s instructions.

Table 2 Selection of genes for qPCR analysis in Brazilian sugarcane cultivars.

Gene	Organism	Gene Bank	Description	Group	Reference	Sequence 5′–3′	
SDH	Saccharum hybrid cultivar Q117	CF576911.1	Succinate dehydrogenase	Organic acids	Casu et al. (2004)	(F) TGCATCACCAAGCTCTTTC	
(R) CCACCTCCAATCATCTTCAC	
MDH	Saccharum hybrid cultivar SP803280	CA119663.1	Malate dehydrogenase	Organic acids	Vettore et al. (2001)	(F) CTTGATGTAATGAGGGCAAATAC	
(R) AGGAGTGGGAGTAATCGTAAG	
SOD	Saccharum hybrid cultivar SP803280	CA148508.1	Superoxide dismutase/Cu-Zn	Oxidative stress	Vettore et al. (2001)	(F) AACCCCGATGGTAAAACACA	
(R) AAGGTGGCAGTTCCATCATC	
POLI			Polyubiquitin	Reference gene	Papini-Terzi et al. (2005)	(F) CCCTCTGGTGTACCTCCATTTG	
		(R) CCGGTCCTTTAAACCAACTCAGT	
Note:

F, Forward; R, Reverse.

RNA quality was confirmed using agarose gel stained with ethidium bromide and visualized under UV light to evaluate the presence of intact 28S and 18S rRNA bands. RNA concentration and purity were measured using a Thermo Scientific NanoDrop 2000 spectrophotometer. Samples were treated with DNase I Amplification Grade (Invitrogen) to remove genomic DNA. DNAse treatment was validated by qPCR using polyubiquitin reference gene primer with the DNAse-treated RNA as a template for reactions. A total of 1.5 µg of DNase-treated RNA was reverse transcribed with a High-Capacity cDNA Reverse Transcription kit (ThermoFisher Scientific) using 100 pmol of oligodTV.

qPCR reactions were conducted using 2 µL of cDNA, 1× Platinum SYBR Green qPCR SuperMix-UDG (Invitrogen), and final primer concentrations of 300 mM for polyubiquitin (reference gene), 100 mM for SDH, 450 mM for MDH and SOD in a final volume of 10 µL. Reactions were conducted in triplicate in an ECO Real-Time PCR system (Illumina) with the following conditions: 2 min at 50 °C, 2 min at 95 °C, [40×] 95 °C for 30 s, 55 °C for 30 s, and 72 °C for 40 s. A melting curve was obtained after each reaction to confirm there were no non-specific amplification products. The used primers are listed in Table 2, and the concentrations were optimized before the reaction efficiency analysis. Reactions had an amplification efficiency between 95 and 105% and R2 > 99%. Data were analyzed using the 2−ΔΔCt method (Livak & Schimittgen, 2001). Observed differences in gene expression were considered significant when p was lower than 0.05 with a 95% confidence interval.

Results

Greenhouse bioassay

Soil incubation with limestone for 50 d resulted in the following chemical parameters: pHCaCl2 = 4.9, potential acidity (H + Al) = 20 mmolc dm−3, exchangeable Al = 1 mmolc dm−3, EB (exchangeable bases) = 22.7 mmolc dm−3, TCEC (total cation exchange capacity) = 74.0 mmolc dm−3, V% (base saturation) = 53 and m% (Al saturation) = 4.22. Aluminum chloride was added to soil samples to induce stress by Al and the following results were obtained: pHCaCl2 = 3.7, H + Al = 58 mmolc dm−3, Al = 21 mmolc dm−3, EB = 16 mmolc dm−3, TCEC = 42.7 mmolc dm−3, V% = 22 and m% = 56.76.

Nutrient content in roots and leaves

The effect of Al and the influence of sugarcane cultivar were evaluated for roots and leaves separately. All results of leaf and root nutrient content significantly different by ANOVA F test (p < 0.05) were submitted to the Tukey’s test (p < 0.05) (Fig. 1). Interaction between sugarcane cultivars and Al levels was highly significant (p < 0.01) for the content of N, P, K, Mg, and S in the roots, and the content of N, P, and K in the leaves. The content of primary macronutrients (N, P, and K) in the plant tissue was highly dependent on the interaction between cultivars and levels of Al. Calcium content in the roots, and Ca and Mg content in leaves were highly significant (p < 0.01) for both isolated factors (p < 0.01) while S content was only attributed to the effect of the cultivars (p < 0.01).

Figure 1 Roots and leaves mean content of primary and secondary macronutrients in sugarcane cultivars cultivated in Dystrophic Typic Hapludox at different levels of Al stress.

(control: soil Al content = 1 mmolc dm−3 and Al saturation m% = 4.22; induced Al stress: soil Al content = 21 mmolc dm−3 and Al saturation m% = 56.76). Significant results for the interaction between sugarcane cultivars and Al levels (ANOVA F test; p < 0.05): (A) Nitrogen (N) contents in roots and leaves; (B) Phosphorus (P) in roots and leaves; (C) Potassium (K) in roots and leaves; (D) Magnesium (Mg) in roots; (E) Sulfur (S) in roots; significant results only for sugarcane cultivars (ANOVA F test; p < 0.05): (F) Calcium (Ca) in roots and leaves, and Mg and S in leaves. The standard error (SE) is represented in the figure by error bars attached to each column. Lowercase compares treatments for each cultivar and uppercase compares varieties for each treatment according to the Tukey’s test, p < 0.05; DW: dry weight.

The range of variation for nutrient content (g kg−1) in the roots were as follows: (i) control treatment-N 5.75–13.50; P 0.29–0.82; K 6.99–24.46; Ca 2.29–3.95; Mg 1.76–3.91; S 1.92–5.59; (ii) Al induced stress treatment-N 6.00–10.50; P 0.34–0.61; K 9.32–12.81; Ca 1.39–3.22; Mg 0.66–1.18; S 1.17–2.11. The nutrient content (g kg−1) in the leaves varied as follows: (i) control treatment-N 13.93–18.91; P 0.90–1.07; K 8.15–13.12; Ca 3.54–5.74; Mg 1.80–2.73; S 1.17–2.10; (ii) Al induced stress treatment-N 14.10–17.33; P 0.77–1.24; K 8.15–11.10; Ca 3.10–4.35; Mg 1.53–2.19; S 1.15–1.78.

Nutrient levels in the plant tissue of cultivars cultivated in the control treatment were higher in most cases. RB928064 presented the greatest content of nutrients in the leaves and in the roots, regardless of treatment with Al, confirming its high nutritional requirement (RIDESA, 2010). Cultivars RB928064 and RB935744 showed greater nutrient absorption when compared with RB867515, but they showed a decrease in the nutrient content after exposure to stress by Al. Even with a lower nutrient uptake (intrinsic characteristic of the cultivar) compared to other cultivars, RB867515 showed a different behavior, with a higher content of N, P, and K in induced Al stress treatment when compared to the control (Fig. 1).

Al content in roots and leaves

Interaction between the factors sugarcane cultivar and Al level was not statistically significant to explain Al content in roots or leaves. Aluminum content in the roots was highly significant (p < 0.01) for both isolated factors. The levels of Al in leaves were only explained by the cultivar (p < 0.01) (Fig. 2).

Figure 2 Aluminum (Al) mean content in roots and leaves of sugarcane cultivars cultivated in Dystrophic Typic Hapludox at different levels of Al stress.

(control: soil Al content = 1 mmolc dm−3 and Al saturation m% = 4.22; induced Al stress: soil Al content = 21 mmolc dm−3 and Al saturation m% = 56.76); significant results only for sugarcane cultivars (ANOVA F test; p < 0.05). The standard error (SE) is represented in the figure by error bars attached to each column. Uppercase letters compare Al content among cultivars for the same material (root or leaves) according to Tukey’s test, p < 0.05.

The ranges of variation for Al content (g kg−1) in the roots were as follows: (i) control treatment—from 418.57 (RB867515) to 433.07 (RB935744); (ii) Al induced stress treatment—from 426.91 (RB928064) to 544.95 (RB867515). The Al content (g kg−1) in the leaves varied as follows: (i) control treatment-from 23.00 (RB867515) to 87.825 (RB928064); (ii) Al induced stress treatment-from 26.86 (RB867515) to 49.13 (RB928064).

The cultivar RB928064 presented the lowest Al content in the roots, followed by RB935744 and RB867515. There was no significant difference among them. Al content in the leaves and in the root did not follow the same trend. RB928064 had the highest Al content in the leaves compared to the other cultivars. Considering the cumulative total amount of Al in plant tissues (leaves + roots), RB928064 had the lowest content compared to the other cultivars (Fig. 2).

Physiological parameters

Photosynthesis rate was only explained by the cultivar (p < 0.01). Interaction between cultivars and Al levels explained rate of transpiration and stomatal conductance (p < 0.05). RB928064 and RB867515 showed nearly twice the accumulation of CO2 efficiency compared with RB935744. RB928064 showed the highest rate of transpiration and stomatal conductance when in control conditions. On the other hand, it presented lower rates of transpiration and conductance compared to the other studied cultivars when subjected to induced Al stress treatment (Fig. 3).

Figure 3 Physiological parameters (mean values) analyzed in leaves of sugarcane cultivars cultivated in Dystrophic Typic Hapludox at different levels of Al stress.

(control: soil Al content = 1 mmolc dm−3 and Al saturation m% = 4.22; induced Al stress: soil Al content = 21 mmolc dm−3 and Al saturation m% = 56.76); significant results for the interaction between sugarcane cultivars and Al levels (ANOVA F test; p < 0.05): (A) Stomatal conductance; (B) Transpiration; significant results only for sugarcane cultivars (ANOVA F test; p < 0.05): (C) Photosynthetic rate. The standard error (SE) is represented in the figure by error bars attached to each column. Lowercase compares treatments for each cultivar and uppercase compares cultivars for each treatment according to Tukey’s test, p < 0.05.

Interaction between cultivar and Al level explained the content of chlorophyll a (p < 0.05), chlorophyll b (p < 0.01), and total chlorophyll (a + b) (p < 0.01). In the control treatment, RB867515 had higher content of chlorophyll a while RB935744 and RB928064 cultivars had higher levels of chlorophyll b and a + b. The addition of Al did not cause a difference in the content of chlorophyll a, b or a + b among cultivars. Aluminum induced treatment resulted in a decrease of almost 50% in the content of chlorophyll a for RB867515 and chlorophyll b content for RB867515 and RB935744 (Fig. 4). Regarding the content of chlorophyll b and a + b, the stress induced by Al caused an increase in RB928064 and a decrease in RB935744 cultivars, but did not cause changes in RB867515 (Fig. 4).

Figure 4 Chlorophylls mean content in leaves of sugarcane cultivars cultivated in Dystrophic Typic Hapludox at different levels of Al stress.

(control: soil Al content = 1 mmolc dm−3 and Al saturation m% = 4.22; induced Al stress: soil Al content = 21 mmolc dm−3 and Al saturation m% = 56.76). Significant results for the interaction between sugarcane cultivars and Al levels (ANOVA F test; p < 0.05). The standard error (SE) is represented in the figure by error bars attached to each column Lowercase compares treatments for each cultivar for each type of chlorophyll and uppercase compares varieties for each type of chlorophyll according to Tukey’s test, p < 0.05; FW: fresh weight.

Gene expression analysis

A statistically significant variation (t test, p < 0.05) in the expression of the SOD, MDH and SDH genes was observed with exposure of the cultivars to stress by Al (Fig. 5). The only change in the SDH gene expression was observed in the roots of the cultivar RB928064, which decreased (0.35-fold) after the stress induced by Al (Fig. 5A). The MDH gene expression decreased (0.24-fold) in the roots of cultivar RB935744 (Fig. 5B), and in the leaves (0.13-fold) of cultivar RB867515 (Fig. 5C). A high MDH gene expression (3.35-fold) was observed in the roots of RB867515 (Fig. 5C). The SOD gene expression was significant in the three cultivars, increasing equally in leaves (1.21-fold) and in roots (1.31) of cultivar RB928064 (Fig. 5A). In cultivar RB935744, greater SOD gene expression (2.74-fold) was observed in the leaves. In the roots, gene expression decreased (0.55-fold) (Fig. 5B). The inverse behavior was noticed for the cultivar RB867515, in which there was a decrease in the SOD gene expression (0.14-fold) in the leaves, and an increase (2.36) in the roots (Fig. 5C).

Figure 5 Expression of genes of sugarcane cultivars in nutrient solution at different levels of Al stress (0 and 3,000 µmol L−1).

Expression of genes related to Al tolerance in plants submitted to Al-treatment in comparison with the control plants. Polyubiquitin gene was used as a reference gene. Bars represents the average of relative expression of SDH, MDH and SOD of each cultivar (the average was composed of the appropriate biological duplicates and triplicates of the qPCR reaction). The error bars were calculated according to Livak & Schimittgen (2001). Asterisks indicate significant values (t test, p < 0.05; 95% prediction interval).

Discussion

Soil incubation with limestone for 50 d was enough to achieve proper chemical conditions for sugarcane cultivation: V% close to 60 and low levels of exchangeable Al (1 mmolc dm−3) and m% (4.22%). Soils with great fertility problems have very high exchangeable Al content (higher than 3 mmolc dm3), as well as Al saturation (m%) higher than 50%. The m% (56.76%) and exchangeable Al (21 mmolc dm−3) values were high enough to guarantee a sufficient level of Al stress for the sugarcane cultivars to verify their behavior in response to high levels of Al in low-fertility soil (Table 1).

The treatment induced by Al caused a change in nutrient content in the plant tissues of the three cultivars. In root tissues, N, P, Ca, and Mg concentrations were higher for the cultivar RB928064, while K and S content were higher for RB935744. The lowest levels of N and Mg were observed in the roots of the RB935744. Low root P, K, Ca, and Mg content was also observed in RB967515 (Fig. 1). While Al decreased the absorption of nutrients by RB928064 and RB935744, there was an increase in N, P, and K root content in RB967515 under Al stress. RB928064 exhibited the highest K, Ca, Mg, and S leaf content, and the lowest N and P content. The highest content of N, P, and Mg and the lowest content of K were observed in the leaves of RB935744. The highest nutrient content of cultivars RB928064 and RB935744 are probably due to the intrinsic characteristic of these cultivars as they are recommended in agricultural environments with greater soil fertility potential (RIDESA, 2010). The lowest content of N, K, Ca, and Mg were observed in the leaves of RB967515. Despite having the highest Al (Fig. 2) and the lowest nutrient content in the tissues (Fig. 1), RB867515 showed greater resilience in nutrient content when submitted to Al stress (Figs. 1A–1C). Its rusticity, recognized by its breeding program, allows its cultivation in more restrictive agricultural areas, and development in less fertile soils (RIDESA, 2010). Oliveira et al. (2010) demonstrated the wide variation in nutritional requirement for the production of one ton of stalk per hectare (TSH) in the irrigated cane-cycle of 11 Brazilian sugarcane cultivars. Sugarcane TSH ranged from 155 to 256 t ha−1, and wide nutrient extraction was observed (0.53-1.27 kg t−1 of N; 0.10–0.17 kg t−1 of P; 0.83–2.58 kg t−1 of K; 0.92–1.52 kg t−1 of Ca; 0.35–0.50 kg t−1 of Mg). Maintaining or increasing nutrient absorption under stress by Al is a desired characteristic in sugarcane cultivars, since it contributes to lower the occurrence of damage to the cells and to the plant as a whole. The first Al contact with the plant occurs at the cell wall, where it is absorbed by mass flow, being mainly accumulated in the apoplast. The binding of Al3+ to the plasma membrane alters its negative surface. This induces depolarization, especially at the root apex, the most sensitive zone to Al as well as the most active (Gupta, Gaurav & Kumar, 2013; Rao et al., 2016; Ryan, 2018; Yadav et al., 2020). These changes in plasma membrane properties affect both ion absorption and transport (Horst, Wang & Eticha, 2010). Water and nutrients do not enter the cell with altered permeability, and the plant becomes more susceptible to drought and nutritional stress. This was observed in cells of maize roots subjected to Al, where changes in the cellular integrity of the root apex resulted in lower Ca, Mg, and K absorption (Wang et al., 2015). The decrease in the nutrient content in leaves and roots after exposure to Al was observed for the three sugarcane cultivars (Fig. 1). There was an exception in the cultivar RB867515, which showed an increase in the root P and K content in response to stress by Al.

The Al content in the roots and leaves was consistent with previous reports for sugarcane (Watt, 2003). The cultivar RB928064 presented the lowest Al content in the roots, followed by RB935744 and RB867515, which did not differ significantly from each other. Aluminum content in the leaves and in the root did not follow the same trend. RB928064 had the lowest total amount of Al in plant tissues (leaves + roots) compared to other cultivars (Fig. 2) but the highest Al content in the leaves (Fig. 2). The greatest Al leaf content of RB928064 (Fig. 2) can be due to transporter activity that acts on the plasma membrane, removing Al from the negative sites of the root’s apoplast. Aluminum is bound to organic acid, transported by the xylem, and then forms a complex for leaf storage by vacuolar compartmentalization (Kochian et al., 2015; Wang et al., 2015; Bojórquez-Quintal et al., 2017). Aluminum has no phytotoxic effect on the plant once bound to organic acid and compartmentalized in a cell structure by this accumulation mechanism (Kochian et al., 2015; Yan et al., 2021). This decreases the level of toxic Al that would cause cellular rupture and impair root growth (Klug & Horst, 2010). In contrast to species or cultivars that accumulate Al, some plants use a less specialized exclusion mechanism of intraradicular and extracellular compartmentalization (Kochian et al., 2015). Apparently, RB935744 and RB867515 cultivars use this mechanism as suggested by the higher concentrations of Al in the root tissue. More specific studies to identify the metabolic routes of Al detoxification may clarify specialized mechanisms employed by sugarcane cultivars.

Regarding stomatal conductance and transpiration rate, RB935744 and RB967515 cultivars proved unresponsive to induced Al stress. RB928064 has presented higher stomatal conductance and transpiration rate than other cultivars in control conditions, parameters that were reduced by half in the presence of Al (Figs. 3A, 3B). An indirect effect of Al phytotoxicity is the decrease in water use efficiency resulting from damages to the root system. In the absence of chemical (phytotoxicity by Al) or physical (compaction) soil impediments, root system growth of sugarcane under field conditions can exceed 4 m in depth (Laclau & Laclau, 2009), or even 6 m in depth (Smith, Inman-Bamber & Thorburn, 2005). The average root system depth of sugarcane cultivated in the Brazilian Cerrado conditions (savanna) is estimated at 0.6 m due to high soil Al saturation and low levels of Ca, among other factors (Koffler, 1986). The absence of a well-developed and well-functional root system decreases stomatal conductance, transpiration rate, internal CO2 concentration, and photosynthesis rate (Basnayake et al., 2015; Ferreira et al., 2017). The decreased transpiration rate and stomatal conductance of RB928064 (Figs. 3A and 3B) indicates that the plant has increased susceptibility to water stress as a response to the presence of Al. The sensitivity of these parameters was even observed when there was no water deprivation in our study. Aluminum stress tolerant cultivars tend to be more water-efficient and produce more dry matter per gram of transpired water. Even though water stress was not the focus of our study, these results may guide future studies on water use efficiency of sugarcane cultivars.

There was no decrease in content of chlorophyll a, chlorophyll b, and total chlorophyll (a + b) for the RB928064 cultivar treated with Al. The presence of Al greatly increased the content of chlorophyll b and total chlorophyll (a + b) for RB928064. Content of chlorophyll a in the leaves of RB867515 decreased with Al addition (Fig. 4). However, the RB928064 and RB867515 cultivars had higher photosynthesis rates than RB935744 (Fig. 3C), and the decrease in chlorophyll content does not seem to have proportionally affected photosynthesis rate. The rate of CO2 assimilation by plants may vary according to species and even among cultivars, depending on factors such as plant age, Al concentration, and the time of exposure to Al (Fonseca Júnior et al., 2014). Particularly for cultivar RB867515, the lowest chlorophyll content was associated with the lowest foliar levels of N and Mg (Fig. 1F). Chlorophyll molecules are formed by complexes derived from porphyrin, with a central Mg atom. Nitrogen is directly linked to the synthesis of chlorophyll-a that is used in the first stage of photosynthesis (photosystem I) (Taiz & Zeiger, 2001).

Damages from Al in the photosynthetic process of sugarcane plants are associated to Al interference in both energy absorption and electron transfer processes in photosystem II; this component of the photosynthetic complex plays an essential role on the response to environmental stress and when chlorophyll b predominates (Ecco, Santiago & Lima, 2013).

Greater photosynthetic efficiency of RB867515 is probably more closely associated with the content of chlorophyll b, which was not negatively affected by the presence of Al (Fig. 4).

Gene expression analysis showed that the highest MDH gene expression of RB867515 (3.35-fold) was in the roots (Fig 5C). This gene is responsible for the reversible reduction of oxaloacetate to malate. In plants, malate is involved in respiration, nutrient absorption, stomatal movement, nitrogen assimilation, and photosynthesis (Vance & Heicher, 1991). Some studies even discuss the relationship between the production of organic acids (succinic acid, citric acid, malic acid, and oxalic acid) and MDH gene expression. When MDH gene is overexpressed, there is an increase in organic acid exudation, which increases Al tolerance (Tesfaye et al., 2001; Wang et al., 2010). High MDH gene expression suggests organic acids are involved in mechanisms of Al detoxification used by RB867515. The higher Al content in the root tissue (Fig. 2) may have triggered a higher expression of these genes after the absorption of Al by the root. Organic acids act through Al chelation to maintain Al in the root apoplast, in the root symplasm, or in the cell walls of the root. The exposure to stress by Al decreased MDH gene expression in the roots of RB935744 (Fig. 5B), and in the leaves of RB867515 (Fig. 5C). Most studies with Al in plants only analyze the root system, but a differential expression of genes that participate in the Krebs Cycle and are related to the production of organic acids, such as MDH, have been reported in leaves. In the transcriptome analysis, 668 DEGs were found in the leaves, number that is significantly higher than those reported for maize roots (Mattiello et al., 2010).

The SOD gene is related to oxidative stress and showed increased leaf or root expression for all cultivars (Fig. 5). Aluminum is highly reactive and alters the functions of the plasma membrane and interacts with the phospholipid layer, causing peroxidation and an increase in toxic reactive oxygen radicals (ROS) (Panda, Baluska & Matsumoto, 2009). The family of antioxidant superoxide dismutase (SOD) enzymes is the most effective in preventing cell damage in plants caused by oxidative stress; it is present in all cells of all plants (Wang et al., 2016). The cultivar RB928064 showed an equitable increase in SOD gene expression in the roots and leaves (Fig. 5A). For the RB935744 cultivar, there was an increase in leaf SOD expression and a decrease in root SOD expression (Fig. 5B). The opposite pattern was observed for the cultivar RB867515, with a decrease of SOD expression in the leaves and an increase in SOD expression in the roots (Fig. 5C). The effect of gene expression related to SOD is governed by subcellular sites found in tissue cells where oxidative stress is generated (Wang et al., 2016). The literature shows an indistinct behavior of SOD expression in the roots and leaves of several cultivated grasses. In rice, the Al-tolerant Vandana cultivar presented lower ROS production in roots and leaves, as well as increased SOD gene expression in these tissues in response to Al treatment compared to the Al-sensitive cultivar HUR-105 (Bhoomika, Pyngrope & Dubey, 2013). In rye, higher Al accumulation, ROS, and cell death occurred in the Al-sensitive cultivar Riodeva compared to the Al-tolerant Petkus, along with lower SOD gene expression in the roots. Petkus, however, showed greater expression in roots than in leaves (Sánchez-Parra et al., 2015). A similar behavior was observed in ryegrass (Cartes et al., 2012). These results corroborate with data observed in the present study, where variation in the SOD gene expression in tissues within the same species may occur according to the inherent Al tolerance (or sensitivity).

The only change in SDH gene expression was observed in the roots of the cultivar RB928064, which decreased after the stress induced by Al (Fig 5A). The succinate dehydrogenase has a central role in mitochondrial metabolism, acting in the tricarboxylic acid (TCA) cycle and in the aerobic respiratory chain. It catalyzes the oxidation of succinate to fumarate and the reduction of ubiquinone to ubiquinol (Figueroa et al., 2001).

Up-regulation of expression in response to Al stress was observed in root tissues for the SOD gene in RB928064 (Fig. 5A) and for MDH gene in RB867515 (Fig. 5C). In leaf tissues, the same was observed for SOD gene in RB928064 (Fig. 5A) and RB935744 (Fig. 5B). There was a higher frequency of down-regulation of gene expression under induced Al stress, noticed for SDH gene in the root tissue of RB928064 (Fig. 5A), MDH and SOD genes in the root tissues of RB935744 (Fig. 5B), and for MDH and SOD genes in the leaf tissues of RB867515 (Fig. 5C). Recently, Rosa-Santos et al. (2020) reported in RB855453 (Al-sensitive) and CTC-2 (Al-tolerant) cultivars that genes related to the auxin signaling, detoxification, ROS protection, and signal transduction were significantly involved in sugarcane response roots to Al.

Conclusions

The results of this study add to the demand for experiments to determine the role and gene expression related to Al stress in sugarcane. Gene expression of MDH, SDH (both related to the production of organic acids), and SOD (related to oxidative stress) was evaluated by qPCR. Complimentary evaluations for practical application in agronomic purposes have also been proposed, such as changes in nutritional and physiological parameters in response to Al stress. Brazilian sugarcane RB867515, RB928064, and RB935744 cultivars exhibited very different response patterns to stress by Al. Exposure to Al caused up-regulation of gene expression in root (SOD gene in RB928064) and leaf tissues (SOD gene in RB928064, and MDH gene in RB928064 and RB935744). Aluminum stress induced down-regulation of gene expression in both root (SDH gene in RB928064 and MDH and SOD genes in RB935744) and leaf tissues (MDH and SOD genes in RB867515). This investigation provides insights on Al detoxification mechanisms employed by sugarcane cultivars. Regarding gene expression, the cultivar RB867515 uses mechanisms of exclusion of Al by the production of organic acids in the roots while RB928064 seems to employ a mechanism of transport and accumulation of Al in the leaves. The sugarcane cultivar RB867515 was considered more Al-tolerant based on the following results: same content of primary macronutrients (N, P, and K) in the leaves and roots regardless of the level of stress by Al; highest Al content in the roots; lack of effect on rates of photosynthesis, transpiration and stomatal conductance by Al stress; gene expression of both MDH and SOD in the roots. The RB928064 and RB935744 decreased nutrient content in the leaves and roots after exposure to Al and exhibited the lowest Al content in leaves + roots. Aluminum also decreased photosynthesis in RB935744 and the transpiration and stomatal conductance in RB928064. Finally, RB928064 showed only low expression of SOD in roots and leaves, while RB935744 showed important expression of the SOD gene only in leaves. Regarding Al-tolerance, the sugarcane cultivars were classified as the following: RB867515 > RB928064 = RB935744.

These findings may contribute to the selection of potential plants for breeding programs and for the development of cultivars better adapted to acidic soils. In addition, Al-tolerant cultivars enable a more sustainable agricultural system as they have lower requirements for soil input and are less susceptible to water stress due to the full development of the root system.

Supplemental Information

Supplemental Information 1 Raw data from the RT-qPCR assay.

Click here for additional data file.

Supplemental Information 2 Aluminum content in roots and sugarcane leaves grown in greenhouse-treatment with aluminum and control.

Click here for additional data file.

Supplemental Information 3 Sugarcane Physiological Parameters.

Click here for additional data file.

Supplemental Information 4 Sugarcane Nutritional Analysis.

Click here for additional data file.

Supplemental Information 5 Raw data obtained from chlorophyll analysis.

Click here for additional data file.

Additional Information and Declarations

Competing Interests

Author Contributions

Data Availability

The authors declare that they have no competing interests.

Mariane de Souza Oliveira conceived and designed the experiments, performed the experiments, analyzed the data, prepared figures and/or tables, authored or reviewed drafts of the paper, and approved the final draft.

Sâmara Vieira Rocha conceived and designed the experiments, performed the experiments, analyzed the data, authored or reviewed drafts of the paper, and approved the final draft.

Vanessa Karine Schneider conceived and designed the experiments, performed the experiments, analyzed the data, prepared figures and/or tables, authored or reviewed drafts of the paper, and approved the final draft.

Flavio Henrique-Silva conceived and designed the experiments, performed the experiments, authored or reviewed drafts of the paper, supervising all aspects of conducted studies, and approved the final draft.

Marcio Roberto Soares conceived and designed the experiments, performed the experiments, authored or reviewed drafts of the paper, supervising all aspects of conducted studies, and approved the final draft.

Andrea Soares-Costa conceived and designed the experiments, performed the experiments, analyzed the data, authored or reviewed drafts of the paper, contribution in the acquisition of the financial support for the project leading to this publication, and approved the final draft.

The following information was supplied regarding data availability:

Real Time PCR raw data and other measurements are available in the Supplemental Files.

The cultivars RB928064, RB935744, and RB867515 can be found at https://en.ridesaufscar.com.br/variedades.

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
