# Peer review of "Physiological, nutritional, and molecular responses of Brazilian sugarcane cultivars under stress by aluminum"

_PeerJ, doi:10.7717/peerj.11461_

## Round 0.1 · original submission · Major Revisions

Dear Dr. Rocha,

Please find below the suggestions of three independent reviewers, which are unanimous regarding the interest and quality of the work. However, before being accepted for publication, I suggest that you perform the indicated changes, paying particular attention to the language editing (especially compound sentences), clarification of the experimental procedures, and better integration of the results and discussion.

I will be glad to receive the revised version.

Sincerely,
Ana I. Ribeiro-Barros

Reviewer 1 ·

Basic reporting

Structure conform
Only 17% of the literature cited with less then 5 years — some update could be relevant
Figures of good quality

Experimental design

Clear definition of research question, relevant and meaningful
Excessive description of analytical methods — determinations with “method X, according to “citation” were enough for common analytical methods
Sufficient information on raw data and experimental design

Validity of the findings

Some results can be better explained and I am not completely satisfied:
Fig 1.2 — legend said that for all, but F is different from the others
— for A, B, C, D and E, “lower letters to compare each cultivar for the four treatments” but what about roots and leaves? It seems they were compared separately but is not said
— Please confirm all the letters of Tukey’s test
— For F legend should be “means” and not “significant means”
Fig.2 — The only comparison is between roots or between leaves. Why do not include interaction?
— Please confirm all the letters of Tukey’s test
Fig. 5.1 —graphics compare relative variation based on control. It is better clarify legend

Reviewer 2 ·

Basic reporting

i) The article deals with important aspects for the understanding of differences between varieties of sugarcane under aluminum stress. These differences were observed in physiological, nutritional and gene expression parameters. It is an article that is within the scope of the journal;
ii) The article presents a good introduction, with clear objectives;
iii) As a way to improve the introduction, I suggest using a recent study that involves the identification of genes expressed in sugarcane cultivars subjected to aluminum stress. The study “DOI: 10.3390/ijms21217934 (Rosa-Santos et al., 2020)” may provide some data to improve the text contained in lines 116 - 137.

Experimental design

i) Consistent methodology, with good detailing;
ii) I suggest using information on the choice of genes (Introduction, lines 142-144) in the Methodology section;
iii) Appropriate protocols/methods. However, I make two suggestions:
A – The information “Leaf +1 is also called the top visible dewlap (TVD) and is the first uppermost fully expanded leaf from the top of the plant that has a visible dewlap or distinct collar. Leaf +2 is the leaf immediately below leaf +1” (lines 214-216), may perhaps be removed;
B – In the qPCR analysis: I have doubts about the type of root sample. Was the entire root system collected or only certain segments of the root? (Line 301). There may be variations in the molecular response in relation to the effect of Al depending on the root zone.

Validity of the findings

i) There is coherence between the information presented in the Introduction, the objectives, and what was discussed based on the results achieved;
ii) Well-designed discussion, requiring few adjustments;
iii) In my opinion, the Conclusion section can be improved. The information discussed at the end of the Discussion section more coherently addresses the objectives of the study. The Conclusion section of the manuscript conveys information more related to perspectives.
In addition, according to the phrase “…imply in new management strategies that allow to allocate the most tolerant variety in places with greater aluminum toxicity” (Conclusion, lines 566-567), and with the previous information presented in the Results and Discussion, was it clear which variety was the most tolerant? The text on "strategies" is confusing.

Additional comments

i) I suggest standardizing the term “varieties” in the manuscript. Suggested change to the term “cultivar”, for example, title in Table 2 and other locations;
ii) Review the references. Few adjustments, for example, line 608, 675, and 788;
iii) I suggest standardizing the information “RT-qPCR or qPCR”. For example, I suggest changing the title of Table 2 to: "Selected genes for qPCR analysis in varieties of sugarcane";
iv) Review the information/descriptions for all figures. Prioritize standardization. For example: in the Figure 1 legend, “Tukey test at 5% significance” appears twice.


In addition, I would like to ask some questions:
A - Why were these three varieties of sugarcane selected in this study? Are there prior assessments of their tolerance and susceptibility to Al toxicity?
B - Based on the following information “There was observed an increase in the leaf expression of three genes related to Al tolerance, indicating that the Al tolerance mechanism in these sugarcane varieties may be related to higher SDH, MDH and SOD gene expression in the leaves” (Results, lines 391-393), I ask the following questions:
B1: We know that the effects of Al on the root are crucial for plant responses. Couldn't the modulated mechanisms in the roots indicate tolerance responses in these genotypes? There are differences between the 3 sugarcane varieties, especially when we analyze the genes expression profile in the roots of RB867515 variety (Figure 5);
B2: We know that SOD enzyme is essential in the antioxidant defense system in plants, but can the increase in its expression definitely represent a mechanism of tolerance to Al?
The expression of other genes (antioxidant enzymes) is essential to explain how different sugarcane genotypes can neutralize oxidative stress. I understand that the term “indicating” was used, but maybe the phrase can be improved and used in the Discussion section (especially when it was discussed about the role of SOD, lines 505-528).
B3: The information “These results point to a differentiated mechanism of resistance among cultivars, where the plant responds as a whole to Al stress” (Discussion, lines 555-556) does it cause confusion?
Mainly because it has already been reported: “There was observed an increase in the leaf expression of three genes related to Al tolerance, indicating that the Al tolerance mechanism in these sugarcane varieties may be related to higher SDH, MDH and SOD gene expression in the leaves” (Results, lines 391-393).
Summary: I suggest a review of the use of this informations.

Final considerations:
Thank you for the opportunity to contribute to the manuscript.
I believe that the manuscript has clear English, but I am not entirely able to give an accurate opinion on grammar.
I consider it a good manuscript and needs some adjustments.
Best Regards,
Reviewer

Reviewer 3 ·

Basic reporting

The overall quality of English writing should be improved. Some compound sentences, for example, "RB928064 showed a higher transpiration rate and stomatal conductance under control conditions, as well as lower values lower than those of the other varieties in the treatment with Al." should be rewritten into two sentences to deliver a clear message about the results. Please check other compound sentences. The discussions should be concise and consistent with the results. Figures and tables should be appropriately cited in the discussion section when the results were referred to. Many sentences that showed results or discussions from other studies, for example, "Al accumulation is a detoxification mechanism where a chelate is formed between Al and an organic acid", lack appropriate reference citations. Overall writing should be written using a scientific writing style. A good example is a paragraph in line 434-455, where the writing is clear and the discussions were made based on the results obtained from the study. The authors tend to use theories that are remotely relevant to the results of the experiments. So many questions arise in my mind. For example, the sentence in lines 420-425 raised the question of whether the sugarcane used in the experiment was dehydrated and what parameters could be assessed to confirm the water absorption limitation. The figure legends should be carefully written. The legends of figure three did not provide any information about panel C, which showed important results. I feel that the manuscript could be improved with a professional English and scientific wring style.

Experimental design

The research question is not very clear. The authors did not define why they used RB928064 (medium cycle), RB867515 (medium cycle), and RB935744 (late cycle) in the AI treatments. Do these varieties show different Al tolerance levels? The background of these varieties regarding Al tolerance should be provided in the introduction. Many studies usually compared the response of stress-tolerant and stress-sensitive genotypes. The authors may rank these varieties by the Al-tolerant level and focus their discussions on the different responses between plant varieties. I think discussions can be more concise and pertinent.

The methods were clearly described with sufficient detail and information to replicate. The authors may additionally test lipid peroxidation in root to support their discussion about the expression change of SOD gene. ROS generation can be part of the signaling process.

The authors showed that the translocation of Al is one important stress alleviation mechanism. They should test the expression changes of genes that are known to be directly involved in the Al translocation process. In the introduction, the authors mentioned the role of ALS1 and ASL3 genes in such translocation. The expression change of these genes may be tested.

Validity of the findings

The authors should provide complete data. For example, while the stomatal conductance and transpiration rates were compared between control and treated plants, the photosynthetic rates were shown from one condition. The conditions where the photosynthetic rates were obtained were not mentioned. Because the authors mentioned in lines 368-369 that the photosynthetic rates were not affected by the Al treatments, they should provide the results from both conditions. Figure 5 is another example of unclear data presentation. The authors mentioned in lines 380-381 that "The varieties treated with Al of this study presented higher expression of genes related to Al tolerance in comparison with the control (Fig. 5)." In figure 5, the expression of the Polyubiquitin gene was shown as the control. Why did the authors compare the expression of, for example, the SOD gene with the Polyubiquitin gene? The expression of a gene in the sugarcane under treatment conditions should be compared with the expression of the same gene under control conditions. The ambiguous data negatively affects the reliability of the reported results.

Additional comments

All experiments in this study were well performed. However, the interpretation and the presentation of the results are vague. All weak points (the examples above and other similar cases) should be carefully reviewed and corrected throughout the manuscript. The data and the result reports should be carefully checked. The speculations in the discussion should be closely relevant to the results of the study. The authors should focus their discussion on the difference in Al responses between sugarcane varieties rather than the mechanisms by which sugarcane may be used to respond to stress. The expression of genes directly relevant to the Al responses should be investigated. Please add references to all sentences referring to results and knowledge from other studies.

---

## Round 0.2 · Minor Revisions

Dear Dr. Rocha,
It is my pleasure to inform you that your MS - Physiological, nutritional, and molecular responses of Brazilian sugarcane cultivars under stress by aluminum - is almost ready for publication in PeerJ. Nevertheless, I strongly advise you to perform a full language editing before publication.

Also, the Section Editor has asked that you add S.E.M or confidence interval error bars on bar plots.

Thank you for your interest in our journal,
Sincerely,
Ana Ribeiro-Barros

Reviewer 1 ·

Basic reporting

Good

Experimental design

Correct

Validity of the findings

Everything corrected from first version

My only correction is on fig. 1,2,3 and 4, when legend say
"Fig 1 Root and leaves content..."
Shoud say
Fig 1 Root and leaves mean content...

and the some for the other 3 figures

Additional comments

Good work correcting the manuscript

Reviewer 3 ·

Basic reporting

The writing is improved. Figure legends are more clear.

Experimental design

The method is clear. All answers are acceptable, consistent with the scope of work.

Validity of the findings

The concerns are clearly addressed.

Additional comments

The manuscript is greatly improved. All my concerns have been addressed.

---

## Round 0.3 · accepted · Accept

Dear Dr. Rocha

It is my pleasure to inform you that your manuscript - Physiological, nutritional, and molecular responses of Brazilian sugarcane cultivars under stress by aluminum - is now Accepted for publication. Thank you for addressing the entire set of questions raised by the peer-reviews and section editor and contributing to the quality standards of PeerJ.

Sincerely,
Ana I. Ribeiro-Barros